# Technologies of Engagement: How Battery Storage Technologies Shape Householder Participation in Energy Transitions

**Sanneke Kloppenburg** [1,*] ⓘ**, Robin Smale** [1] ⓘ **and Nick Verkade** [2] ⓘ

[1] Wageningen University, Environmental Policy Group, Leeuwenborch, Hollandseweg 1, 6706 KN Wageningen, The Netherlands; robin.smale@wur.nl

[2] Eindhoven University of Technology, School of Industrial Engineering and Innovation Sciences, Room 2.04, PO Box 513, 1600 MB Eindhoven, The Netherlands; n.verkade@tue.nl

* Correspondence: sanneke.kloppenburg@wur.nl

**Abstract:** The transition to a low-carbon energy system goes along with changing roles for citizens in energy production and consumption. In this paper we focus on how residential energy storage technologies can enable householders to contribute to the energy transition. Drawing on literature that understands energy systems as sociotechnical configurations and the theory of 'material participation', we examine how the introduction of home batteries affords new roles and energy practices for householders. We present qualitative findings from interviews with householders and other key stakeholders engaged in using or implementing battery storage at household and community level. Our results point to five emerging storage modes in which householders can play a role: individual energy autonomy; local energy community; smart grid integration; virtual energy community; and electricity market integration. We argue that for householders, these storage modes facilitate new energy practices such as providing grid services, trading, self-consumption, and sharing of energy. Several of the storage modes enable the formation of prosumer collectives and change relationships with other actors in the energy system. We conclude by discussing how householders also face new dependencies on information technologies and intermediary actors to organize the multi-directional energy flows which battery systems unleash. With energy storage projects currently being provider-driven, we argue that more space should be given to experimentation with (mixed modes of) energy storage that both empower householders and communities in the pursuit of their own sustainability aspirations and serve the needs of emerging renewable energy-based energy systems.

**Keywords:** battery storage technologies; energy practices; public participation; householders; socio-technical transitions

## 1. Introduction

In Europe and elsewhere, there is an increase in renewable energy generation at domestic and community level. By installing solar panels, more and more householders are becoming prosumers and take responsibility for the decarbonization of the electricity system. However, for the grid, the uptake of solar poses challenges to the balancing of supply and demand of electricity and to grid management. Solar panels only generate energy during day time, whereas a peak in domestic electricity consumption takes place in the evening. Moreover, there are seasonal differences in the hours of day light and in weather conditions. Storage of renewable energy near to their decentralized sources, at the domestic or local level, is increasingly seen as a solution to this problem. Rapid developments in battery technologies have even led some to claim that we are at the brink of a 'storage revolution' [1] that may change the way householders and institutional actors engage with energy in fundamental

ways. In addition to promises about the potential of storage for decarbonization and decentralization of the energy system, storage features in discourses about the empowerment of householders and communities to take more control over their energy use and become more independent from energy suppliers [2,3].

Despite the view of storage as a potential enabler of the energy transition, not much is known about the role that householders play, or are imagined to play, in energy systems that include distributed storage [4]. Yet, home batteries open up a range of possible roles and practices for householders. They enable householders to store their energy for use at a later time, but are also an important element in enabling new energy practices such as sharing and trading energy. These new energy practices place householders in a different relationship with the energy system and its key actors, such as energy suppliers. For example, the use of residential energy storage can help householders to become (more) autonomous in their energy supply, but domestic storage may also be used for Demand Response to help stabilize the grid [5].

In this article our aim is to explore potential ways in which home batteries can enable householders to become engaged in the transition towards low-carbon energy systems. Departing from the idea that energy systems are socio-technical configurations, we identify different ways in which householders and communities can become involved in low-carbon energy systems with storage. We link this idea of socio-technical configurations, or storage modes, to theories of 'material participation' [6,7] that argue that through everyday interactions with (energy) technologies, people can express concerns and 'intervene' in the energy system. Our theoretical argument then is that the ways in which householders (are enabled to) engage with energy storage technologies in an everyday, practical sense at the same time shape their participation in wider energy systems and their transitions.

In the following sections, we first explain our approach to energy storage as a technology of engagement, and the way we conducted our research. Next, we distinguish five different socio-technical configurations -or storage modes- in which householders can play a role. We identify how each mode affords specific energy practices for householders, such as storing, trading, or exchanging energy, and how the performance of these practices implies a particular distribution of tasks and responsibilities between householders and others energy system actors. In discussing the wider potential implications of these new types of engagements, we reflect on how energy storage may foster new collective energy practices and engagements that challenge our traditional understanding of energy communities, but also how these new energy practices often imply automation and reliance on intermediaries.

## 2. Renewable Energy Technologies as Technologies of Engagement

Literature in science and technology studies (STS) views technologies not just as material objects, but argues that the social and the technical are co-dependent and co-evolving [8]. This field stresses that technology and its social context mutually shape each other. Societal values such as sustainability, and ideas about the roles of users shape the technology, and at the same time technologies are constitutive of the social, in the sense that they actively shape their own context of use. Renewable energy technologies too have been approached as configurations of the technical and the social [9–11]. Walker and Cass use of the term 'mode' to understand renewable energy technologies as configurations of technology and social organization [9]. By social organization they refer to the ways technology is 'utilized and given purpose and meaning' [9]. They distinguish for example the traditional 'public utility mode' from modes that have emerged more recently, such as a the 'private supplier', 'community' and 'household mode'. Walker and Cass seek to understand how different modes embed within them different roles for publics in renewable energy deployment. They characterize these roles in terms of people's spatial proximity to the technology and their level of awareness and active engagement with renewable energy. For example, what they call the 'captive consumer' role entails a consumer who is distanced from the sources of renewable energy generation and consumes green energy passively. In an 'energy producer' role, on the other hand, people own and operate their own green energy generation technologies, for example via solar panels on their roofs, and are necessarily active and aware [9]. This approach

thus recognizes the variety of roles and engagements of publics that emerge in relation to different renewable energy configurations.

While Walker and Cass discuss a wide range of technologies, from micro to macro scale, other studies have characterized and categorized different sociotechnical configurations around one particular technology. For community energy storage, Koirala et al. [2] have identified three configurations, namely shared residential energy storage, shared local energy storage, and shared virtual energy storage. This allows them to analyze the various ways in which local communities can use energy storage. Parra et al. [12] describe four categories defined by scale and application: single home storage, community storage, grid storage, and bulk storage. We take a different approach in basing our categorization of different storage modes on the question of how householders can become involved in and use energy storage.

*Storage Modes and New energy Practices for Householders*

Rather than understanding engagement in terms of general 'roles' for 'publics' or positioning ourselves in emerging research on public perceptions of energy storage [13,14], our aim is to examine what these 'publics', as householders who have installed energy storage devices, can do. In other words, we unpack the roles and forms of engagement by focusing on the (new) *energy practices* that become possible in different storage modes. Here we build on the work of Noortje Marres [6], who calls for an appreciation of everyday material practices as forms of participation. She views people's everyday use of energy technologies such as smart meters, as possibilities for public engagements in environmental issues. As she argues, everyday material actions can enable 'practical or physical interventions in current states of affair' [6]. Such an understanding of 'material participation' acknowledges the ways in which people are engaged in sustainable energy transitions through their everyday practices with household and energy devices.

Building on Marres' work, Throndsen and Ryghaug [15] apply the concept of material participation to assess the character of householder engagement in the case of smart grids. They conclude that householders, as 'material publics', articulate widely ranging (and politically engaged) smart grid enactments. Ryghaug et al. [7] argue the introduction of novel energy technologies in householders' everyday lives, such as solar panels, the electric car, and the smart meter, may create new forms of (materially based) energy citizenship. They give the example of the smart meter that through near real-time measurement and visualization of energy consumption makes energy visible in the household. This may result in the articulation of the issue of energy efficiency, and new forms of (practical) participation such as time-shifting of energy consumption, or replacing existing electric appliances with more efficient ones. The theory of material participation thereby challenges the dominant but narrow understanding of participation as involvement in decision-making. Instead, participation also takes the form of households interacting with energy systems through their everyday use of energy technologies in domestic settings, because in these everyday practices, issues around sustainability and climate change are articulated, and energy decisions are taken [7].

We draw upon the theory of material participation to explore how interactions with home batteries can engage people in the energy systems in different ways. For example, through installing a home battery, people can express their concern for climate change. At the same time, the use of batteries can also make them aware of new issues, such as the rhythms of domestic energy production and consumption and the systemic problem of grid balance. Finally, batteries also enable people to intervene in energy systems in a very concrete and physical sense, because batteries allow the redirecting of energy flows between the household and the wider grid. These examples illustrate energy storage devices as 'objects of participation and engagement' [7] in energy systems. Conceptualizing residential energy storage as a technology of engagement thereby allows us to examine not only how different modes imply different roles and energy practices for householders, but also how each mode at the same time shapes householders' participation in the transition to low-carbon energy systems in distinct ways.

In analyzing which different storage modes are emerging and what forms of engagement they imply, we follow a four-step approach (see Figure 1). Our first step is to examine how storage is viewed as a 'solution' to a particular problem, and whose problem this is (or is made to be). Different problematizations of electricity production and consumption entail specific ways of thinking about storage in home batteries as a solution. Some storage rationalities are more directly linked with householders experiences and practices as solar PV owners, while others start from the problems grid operators face in the context of a changing energy system. Starting from these diverse problem-solution sets we then describe the variety of (new) roles and energy practices for householders that are made available. Next, we analyze the distribution of tasks in these practices. It is important to discuss not only the (new) practices that emerge for householders, but also how and with whom these practices are being carried out, as some of these activities and choices may be delegated to technologies or providers and intermediaries. Finally, we examine the storage modes in relation to the wider energy system (outer circle of the figure). Everyday material practices of storing energy in household batteries enable interventions in the direction and management of (green) energy flows within household and between households, but also in the wider energy infrastructures. As such, these practices represent a rather 'direct' form of engaging with, and potentially reshaping the energy system. Our approach therefore also pays attention to the potential implications on the relationships between householders, providers, and technologies in low-carbon energy systems.

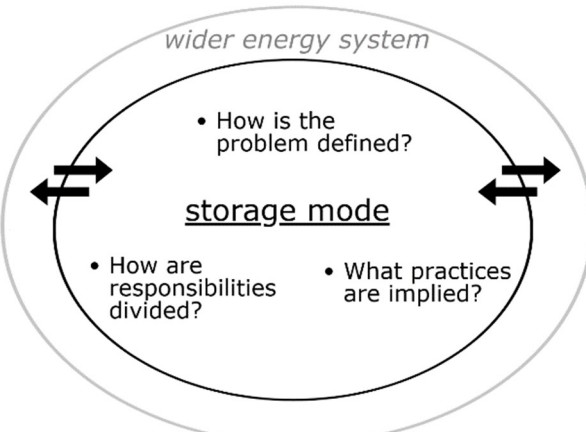

**Figure 1.** Analytical framework for identifying and analyzing storage modes.

## 3. Materials and Methods

This paper builds on empirical data that was collected at different moments and sites in the context of a research project on emerging energy practices in the smart grid (2014–2019). The data are qualitative and consists of interviews with different stakeholders in the energy system in the Netherlands, and to a lesser extent Germany and the United Kingdom. We conducted 14 interviews with providers of home battery systems and services, energy storage experts, NGO's and local governments involved in storage pilot projects. The bulk of the data, however, comes from interviews and observations with householders who were involved in storage pilot projects, or who had installed home batteries themselves. In the fall of 2016, 6 interviews were held with householders in Germany who had installed batteries for individual self-consumption, of which a few also participated in a virtual energy community called SonnenCommunity. In the Netherlands we conducted longer term fieldwork in the context of two demonstration projects. Here 14 interviews were held with householders engaged in the pilot project Jouw Energie Moment ('Your Energy Moment') in which home batteries were used for grid balancing. Furthermore, 30 interviews were conducted in the City-zen pilot project, where householders with batteries engaged in wholesale energy trading. A shortcoming is that we were unable to conduct interviews with local communities who owned and operated storage collectively, because there are relatively few real-world examples of this (but see the Feldheim case reported in [2]).

To gather information about community-owned storage, we therefore relied on interviews with storage providers and document study. Due to the variety of research material and differential access to cases, this research has an exploratory character. Hence, we use the real-world cases to conceptualize and identify the different forms of engagement that the use of home batteries may foster, rather than to systematically evaluate the extent to which new energy practices around storage already result in (new forms of) participation.

## 4. Results

Below we draw out five different socio-technical configurations around home batteries: individual energy autonomy; local energy community; smart grid integration; virtual energy community; and smart grid integration.

### 4.1. Mode 1: Individual Energy Autonomy

In the first mode, Individual energy autonomy (Figure 2), individual households deploy domestic energy storage for the purposes of using (more) self-generated solar energy. The rationality of this mode is optimizing self-consumption of electricity produced by PV panels. Self-consumption itself is a gratifying project for many PV panel owners. As one of the interviewed householders put it, 'I can use the energy, it gives a good feeling to me. To produce it and to use it'. Beyond this, two main motivations are at play here: (long-term) economic reasoning, and desire for autonomy or self-sufficiency. Self-consumption of solar power with domestic storage emerges as an alternative 'business model' for PV owners, as there is a common expectation that in the near future feeding back electricity into the grid will become less financially attractive. Secondly, domestic batteries appeal to householders who wish to become more energy autonomous, and less dependent on subsidies and energy providers. Here, different levels of energy autonomy may be pursued, ranging from going off-grid, to being self-sufficient during a black-out (back-up power), to remaining connected to the grid but relying on it as little as possible. As one of the householders argued: 'Somewhere the subsidies will stop and then you have a big advantage when owning a battery, then you are independent'.

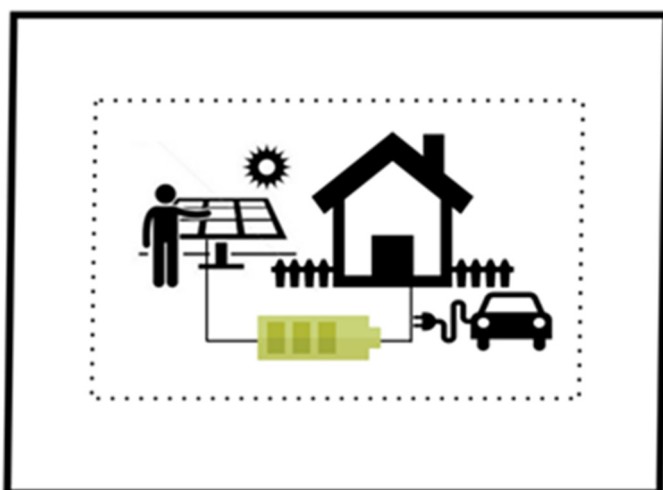

**Figure 2.** Individual energy autonomy.

Home batteries for self-consumption often come along with an app or display on the device itself which enables householders to develop monitoring practices. One of the householders described it as a 'little pleasure' when he uses his app and sees 'that the sun shines and that you can see the battery charging'. Several respondents planned energy-intensive activities, like laundering and dishwashing, in such a way that solar (battery) power is used.

Domestic batteries used for the purpose of enhancing self-consumption place ownership in the hands of householders. However, this does not mean that individual householders can operate their batteries directly. The battery installer can translate the wishes of the householder into the learning algorithms which subsequently govern operation of the battery. As one householder put it: 'With the installer you can configure the battery and optimize everything so that it is attuned to the household. What could the customer do herself? Not so much.'

### 4.2. Mode 2: Local Energy Community

In the local energy community mode (Figure 3), both problem and solution are defined at the community level or within a local area. Local communities cannot always use their locally produced energy within the community itself. For distribution system operators (DSOs), the renewable energy generated by 'green communities' places local pressure on the distribution grid. To both communities and DSOs, an attractive solution is optimizing the local use of locally produced renewable energy. In terms of infrastructures, this mode can either consist of a local community connected to a larger 'neighborhood battery' or be formed by connecting distributed domestic batteries in a local setting. This mode comprises a range of variants from fully self-sufficient off-grid communities to local communities who are sharing energy via the public grid.

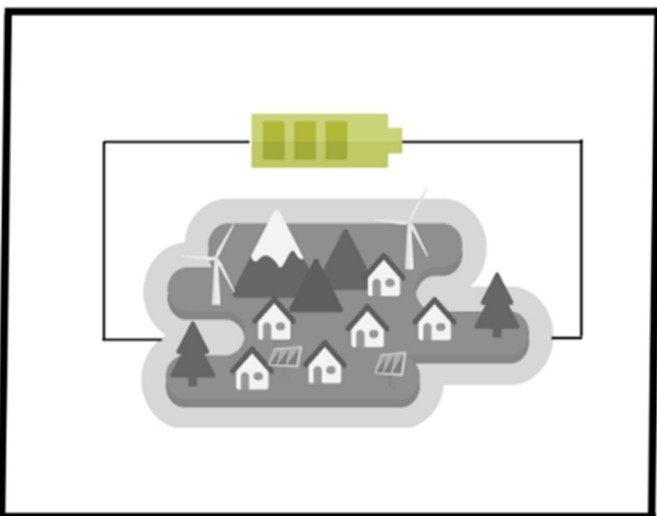

**Figure 3.** Local energy community.

In the local energy community mode, householders become prosumers who not only generate and consume individually, but also for and from the community's pool of energy. This allows for engaging with energy as a 'common good', or a 'common pool resource' [16]. Managing the 'common energy pool' at the community level implies new practices which include the monitoring of not only individual but also community-wide demand and generation; timing-of-demand to match local renewable energy availability (in storage); and energy sharing or peer-to-peer trading between community members.

Theoretically, local energy community storage can be organized in various ways. The local energy community may consist of a pre-existing energy cooperative that decides to add storage to its local renewable energy generation. In the pilot projects we studied, however, the batteries were owned, operated and controlled by other parties than the community itself, requiring little involvement of communities and households. Community energy storage with batteries in its present phase is still experimental, taking place in pilots and living labs. One of the reasons for the absence of 'commercial' variants of this mode are the regulatory barriers to peer-to-peer trading within a community, and to energy collectives becoming their own supplier [12]. In the Netherlands and the United Kingdom, however, regulatory sandboxes are now in place that enable the first communities to experiment with peer-to-peer supply [17]. In conclusion, community energy storage in principle offers

a range of possibilities to organize energy supply and demand at decentral level. Different forms of (community) co-ownership of storage technologies (and generation units) can be imagined, as well as partnerships between energy suppliers and cooperatives; for example, energy suppliers could partner with cooperatives to supply the deficit at moments when the community's energy demand is higher than supply.

### 4.3. Mode 3: Smart Grid Integration

The smart grid integration (Figure 4) mode centers on the increasing problems grid management faces with the ongoing growth of renewable generation at the domestic scale. Grid assets at this scale are not necessarily suited for greater and volatile flows to and from the household. This can be accommodated by making more intelligent use of the grid assets and domestic devices in place with the help of IT, which is the 'hype' [18] called the smart grid. In the smart grid, the demand of households is no longer something that is simply predicted and accommodated by the grid; demand becomes something to be managed and steered at level of the individual household. The flexibility of domestic energy usage becomes an asset to be maximally unlocked and used towards efficient grid management. Domestic energy storage capacity is an ideal flexibility tool from the point of view of the DSO: storage can buffer peaks and troughs in domestic energy demand without requiring the involvement of householders or interfering in their energy use. The rationale of this mode is therefore to align the workings of the batteries (and other household appliances) with the needs of the grid.

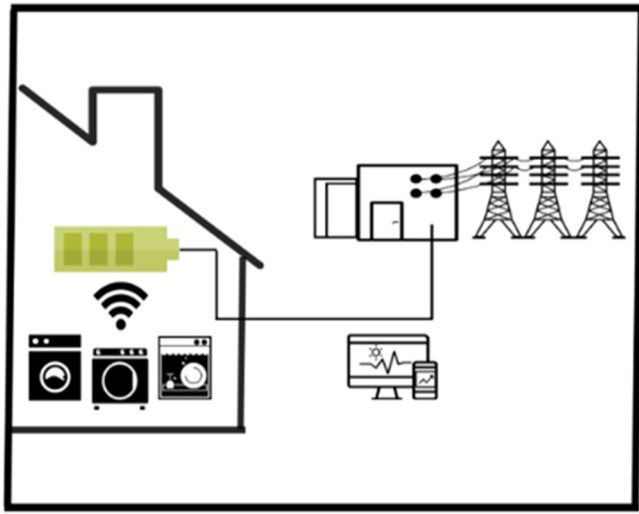

**Figure 4.** Smart grid integration.

Within the smart grid, householders are assigned a role as (active or passive) micro-managers with some responsibility to manage the impact they have on the grid. While they might actively shift some energy usage in reaction to more variable grid tariffs, the smart home with battery storage can also automate some of these decisions. Householders thus 'share' their batteries with the grid, allowing external control of the (dis)charging the battery.

As a result of automation and external control batteries may end up as black boxes, obfuscating the flows of renewable energy in the home and thereby creating a number of new uncertainties for householders. In smart grid pilot project Jouw Energie Moment, many participants critiqued the unintuitive information they were provided with: 'The only thing we pick up on with respect to that battery, is when it is 'humming', which means it is doing something.' The batteries would seemingly switch randomly switch between charging, discharging and neutral, never reaching full charge. Another householder stated: 'I just have no clue of what does what. And whether or not the battery is providing us any benefits.' In this respect, many householders stated that 'naturally,

one would preferably want to be self-sufficient'. However, they were unclear if the batteries were contributing to this objective.

Since DSOs are barred from fulfilling "market-able" roles, the batteries are most likely controlled by an intermediate market actor like an aggregator. Domestic storage and other 'smart appliances' in the home thus become tools for grid supporting services. If householder insight into the functioning of home batteries (and other smart energy technologies) is insufficient, householders may come to see them as external or even invasive tools for solving others' problems [19]: 'At the moment it feels as if I help to solve a logistical problem for the project. I have found space in my home for someone else's experiments. But if I benefit... how can I see that? In effect I can't. I only see a big battery and hear a humming sound.'

### 4.4. Mode 4: Virtual Energy Community

The fourth mode –virtual energy community (Figure 5)- has parallels with the local energy community mode. The situation in which householders possess a battery system to increase their individual self-sufficiency while still relying on conventional energy suppliers to cover additional needs is seen as unsatisfactory. The rationale therefore is to link householders and optimize the use of self-produced energy within the community. While in the local energy community mode members live in the same local area, the virtual community members consist of geographically dispersed households. The members' energy devices (including solar panels, storage devices) are connected via smart meter technologies to a digital platform that allows for the monitoring and exchange of surplus energy. The first real world applications are now emerging (e.g., SonnenCommunity, Schwarmdirigent). One of these virtual energy communities, established by a battery storage provider, is presented as 'a community of [battery owners] who are committed to a cleaner and fairer energy future'. The same provider states that 'as a [member of our community], you don't need your conventional energy provider anymore—you are independent' [20]. In these framings, householders become not only prosumers in a virtual energy community, but also 'part of the energy future'. The goal of the virtual energy community is to meet the energy demand of the community with energy that is generated by the community itself.

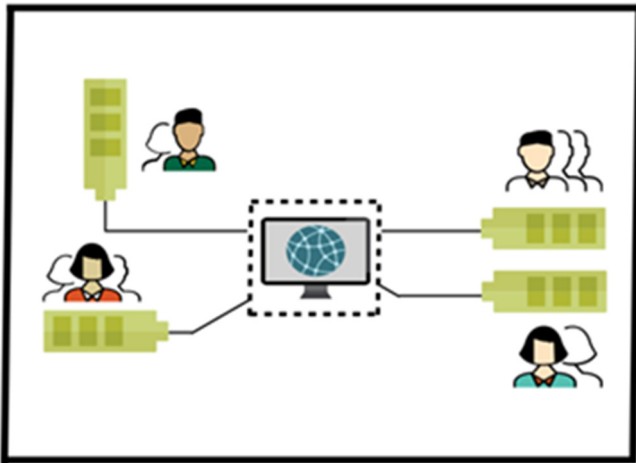

**Figure 5.** Virtual energy community.

In this mode, ownership of the battery is with the individual householders, but the solar surplus that is produced when the batteries are full and/or the stored energy is 'shared' with others. It is important to note here that the sharing or exchanging of energy is virtual: The network does not consist of separate cables between members, but of a digital platform that enables virtual exchange via the existing grid. The meaning of 'sharing' therefore is complex. As one interviewed virtual community member put it: 'the idea is good. With [my friends] I spoke about it, they are part of it. Then I said,

when there's sun at your place, I'm using your power. It's certainly a good idea, as the solar power that is stored, that is too much, can also be used on a place where it rains. But it's all virtual, it's not physical. The energy does not move from one place to the other, but okay, it doesn't matter'.

In the examples we studied, households were not actively engaged in energy exchange in the sense that they needed to decide on when and with whom to enter transactions; the process was managed by a third party—the aggregator—and often highly automatized. It is the responsibility of the aggregator to make sure that the demand within the virtual community matches the supply, so choices and decisions about the distribution of energy are made by this intermediary actor. The exchange of energy is not disclosed or made actionable to householders in the sense that they get insight in for example the current availability of community energy or get rewarded or sanctioned for their energy behavior. What is requested from households is to provide access to their energy data: the energy production, consumption and storage practices of members are monitored, and together with weather forecasts, used to make predictions of supply and demand in the community.

### 4.5. Mode 5: Electricity Market Integration

In the fifth and final mode, electricity market integration (Figure 6), the problem is defined in economic terms: due to competition on free electricity markets and growing renewable energy generation, electricity markets have become increasingly volatile. Batteries allow people to exploit this volatility, because the electricity flow can be temporarily halted, captured, and released again at a later point in time. The rationale of this mode is to align the workings of the batteries with energy market demands in order to create financial benefits for battery owners. In our research, we did not find any commercial variants of this mode yet, but there are examples of trials such as the Dutch pilot project City-zen. The households with batteries do not trade individually because the capacity an individual household can have available is too small. Instead, the participating households are aggregated to form a collective of householders. The aggregator in the Dutch project uses a Virtual Power Plant as the underlying technical infrastructure and explained that 'with all 50 participants, we want to create a large community. This community will be seen as one energy producing or consuming unit' [21]. In the project, the batteries loaded from the grid when prices were low and exported the electricity to the grid when prices were high. Energy thereby became a (tradeable) commodity and householders were ascribed a role as an economic actor who 'acts' according to market rhythms and logics. In the City-zen project, it appeared that for many householders this role as a market actor was at tension with their initial motivation to acquire solar panels for environmental reasons. As one householder explained: 'I didn't first go green with these things to now only think about money!'

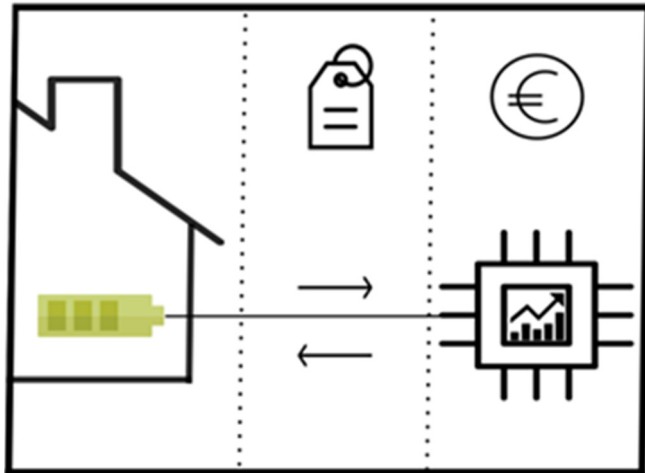

**Figure 6.** Electricity Market Integration.

In theory, in this mode the batteries could be owned by householders as well as third parties. The householders provide (stored) energy, their energy data, as well as the control over the charging and discharging of the battery to an intermediary party in exchange for a monetary reward. The intermediary acts as an aggregator of a group of households and trades on their behalf, by using historical and real-time energy consumption and production data from households in order to make accurate predictions of the amount of energy each household has available for trading. Householders thus engage in trading but this activity does not require specific skills or competences from them, nor does it require or stimulate them to actively adjust their energy consumption practices.

## 5. Discussion

### 5.1. Comparing the Five Storage Modes

Our identification of storage modes shows that a variety of different combinations of home battery storage technology and social organization is currently emerging. In addition to the already more established individual energy autonomy mode, providers are developing new modes that enable energy sharing and providing energy services to the energy system. The five modes we have distinguished differently engage householders in energy production and consumption through storage, in terms of the practices householders are enabled to engage in, and with regard to their relations with the conventional energy system and other householders (see Table 1).

**Table 1.** Five modes of energy storage, including the real-world examples in which fieldwork was conducted.

| Storage Mode | | Householder Engagement | | Real-World Example |
|---|---|---|---|---|
| Modes | Energy Practices | Relation to Conventional Energy System | Engagement Level | Title |
| *Individual energy autonomy* | Self-consumption | Autonomous | Individual | Sonnenbatterie (DE) |
| *Local energy community* | Self-consumption and sharing | Autonomous | Collective | project ERIC * (UK), SWELL * (UK) |
| *Smart grid integration* | Providing grid services (and possibly self-consumption) | Integrated | Individual/collective | Jouw Energie Moment (NL) |
| *Virtual energy community* | Self-consumption and sharing | Autonomous/Integrated | Collective | SonnenCommunity (DE) |
| *Market integration* | Trading (and possibly self-consumption) | Integrated | Individual/collective | City-zen (NL) |

*: no interviews with householders.

First, each mode affords particular energy practices for householders to engage in. In the individual energy autonomy mode, householders engage in self-consumption of stored energy within their household. In the other four modes, self-consumption is complemented with energy sharing, providing grid services, and trading.

Second, the modes entail particular relationships of householders to the conventional energy system. In the individual energy autonomy and local energy community modes, the aim is to increase self-sufficiency at household or community level, and in the ultimate case create a local microgrid. This idea of storage facilitating greater energy autonomy is opposite to the logic of integration that underpins the smart grid and market integration modes. In the latter modes, householders provide energy and services to actors within the energy system and thereby engage in the management of the energy system. The virtual energy community mode is less straightforward to characterize, as it fosters both autonomy and integration. While virtual energy communities may aim at autonomy from conventional energy suppliers, their geographically distributed character means that they need to rely on the public grid for sharing energy.

Third, the five storage modes also imply different types of relationships with other householders. The individual energy autonomy mode is the only mode in which householders do not engage with other householders. The two community modes (mode 2 and 4) connect householders based on shared local identity or values, in order to exchange energy among each other. The market and grid modes, on the other hand, may also aggregate individual households, but these 'collectives' engage in energy transactions with market and grid actors. For householders it may feel as if they participate on an individual basis, while in fact an aggregator treats multiple households as a pool in order to enable their participation in grid management and energy markets [22].

In the remainder of this paper, we want to draw out two important potential implications of these storage modes. Rather than discussing the implications of each mode separately, we reflect on two overarching effects that we consider to bring the most fundamental changes to how people can take part in the energy transition. First, some of the modes enable householders to engage in energy production, consumption and storage via *new collectivities* that challenge our traditional understanding of energy communities. Second, in all of the modes, a large part of the organizational 'work' around storage is performed by *intermediaries and smart technologies*, which challenges the idea of empowerment of prosumers and communities.

*5.2. New Collective Material Practices*

The individual energy autonomy mode is the only mode in which householders produce, store and consume energy within the bounded spaces of their own home. The other four modes comprise material practices which enable householders to form larger collectives and share their hardware and/or energy with others. Such material practices allow householders to go beyond optimizing self-consumption and exchange energy with other households or start transacting with the market or the grid. Existing local energy communities can add batteries to their renewable generation to boost local energy autonomy, but batteries can also enable the formation of new collectives of prosumers. These new collectives are a result of technical infrastructures that interconnect multiple households with batteries. Since aggregation does not require geographical proximity of the households, such new collectives can have members nation-wide as the example of the SonnenCommunity showed. The storage modes that afford collective material practices thereby bring along a range of questions about the character, aims and ideologies of these practices, and how they may and may not differ from the well-known local renewable energy generating communities.

In the literature, a common way to describe renewable energy communities is as 'those projects where communities (of place or interest) exhibit a high degree of ownership and control in renewable energy production as well as benefiting collectively from the outcomes' [23]. Such communities for example consist of local energy cooperatives that develop collective energy practices [24], such as collectively generating solar energy for local use. The aggregation of domestic batteries in particular affords new communities of interest, with new collective practices, to be formed. While the SonnenCommunity is an example of the creation of a community of like-minded users aiming at autonomy from conventional suppliers, other prosumer collectives may align their collective practices with market or grid rationalities. So just like local communities, the new collectives may be oriented towards social goals (e.g., autonomy), sustainability (green energy), and economic goals (profit seeking). An important difference is that the prosumer collectives that are now emerging are often not initiated bottom-up by citizens, but by grid operators, energy suppliers, and start-ups which have the expertise to build and manage the complex underlying technical infrastructure.

How householders can engage these new collectives may differ widely. There are prosumer collectives in which householders participate without being aware of the other 'members', for example when householders are aggregated to provide grid services. In other collectives the connections with other households are made visible in particular ways. For the SonnenCommunity, for example, the provider visualizes the location of community members on a map and shows which type of energy they generate for the community (solar, biogas). In some peer-to-peer exchange platforms

consumers can even choose the peer they want to buy energy from. Emerging prosumer collectives thus shape new collectives which can take very different forms: from the aggregation of householders in collectives that remain invisible and anonymous, to a community of interest with 'members' or 'peers'. An important remaining question, however, is how inclusive these new collectives are for different types of households including lower-income households or tenants. As Ryghaug et al. [7] also argue for the case of electric vehicles and solar panels, the costs of these storage devices may mean that material participation via batteries is not equally accessible to all groups in society.

### 5.3. The Growing Power of Aggregators and Algorithms in New Material Energy Practices

Even though storage devices are located in households or communities, the role of householders in energy storage cannot simply be characterized as the active and aware prosumer. Most of the 'work' around energy storage is carried out by or on behalf of professionals, such as the installation and maintenance of the battery system, the monitoring and management of the battery charging strategy, and the managing of aggregated batteries. The emerging material practices surrounding storage are organized by intermediaries [25] as well as by information technologies.

Intermediary organizations, such as aggregators and green energy suppliers, play a key role in facilitating what householders can do with storage, as well as how, and with whom. Intermediaries are new players in the energy system, who act as a mediator or broker between householders and energy providers. They collectivize householders' energy consumption and production practices and enable and manage their participation in local and national energy systems. In the case of energy exchange among householders, intermediaries may arrange the balancing of supply and demand in the community. Intermediaries thus broaden the options for householders to enter into transactions with other householders and the energy system: transactions that are either too complex, or otherwise inaccessible to (individual) households. For geographical and virtual energy communities who want to become (more) self-sufficient, increased autonomy may thus go along with new forms of dependence on intermediaries who arrange the management and operation of energy exchange. There are concerns about the extent to which householders are able to access the full market potential of their batteries, as business models offered by intermediaries may distribute burdens and revenues unfavorably [26]. Material participation by householders through the purchase of storage batteries is, in other words, not synonymous with householder empowerment.

Information technologies too are a major factor in the management and control of (networked) households with batteries. Smart metering technologies monitor householders' energy consumption, production and storage practices. Hence, it is through these technologies that the householders' energy behavior becomes visible and gets embedded in battery management. Battery charging and discharging strategies often rely on algorithms that predict a household's energy behavior based on its historical energy production and consumption data. In addition, algorithms instruct the direction of energy flows (e.g., discharge to the household, or to the grid). Algorithms may also prioritize certain types of energy (green energy, cheap energy) in the way the battery systems work. In other words, they decide which energy is allowed to flow where and when. Householders choose these 'settings' when they buy a particular storage product or service, and may fine-tune them when the battery is installed. After that, the charging and discharging processes are often automated and users have little possibilities to change settings. Information technologies thus appear as a key factor in enabling connections between local or geographically distributed households and connections with wider infrastructures such as electricity markets. In shaping which transactions can take place, how, and between which entities, digital platforms [27] are becoming a new underlying structure for organizing energy production and consumption at decentral level, with as yet unknown implications for power relations in the energy system [28].

## 6. Conclusions

In this paper, we discussed energy storage as a 'technology of engagement' to better understand how householders and communities through their interactions with storage technologies engage in energy transitions. Drawing on Walker and Cass, we developed the concept of 'storage mode' to examine how battery technologies can be part of diverse sociotechnical configurations. We identified the emergence of five different storage modes, which demonstrates that renewable energy storage can entail a wide variety of relationships and interactions between householders and other energy system actors. To further unpack the various roles and engagements for householders, we examined the problem definitions, practices and task divisions in the modes. Our approach highlights that people can relate to renewable energy technologies not just as supporters or protestors or users, but through a diversity of roles that actively integrate them in the wider energy system (see also [15]): as co-manager or market actor, and as communities or individuals organizing energy production and consumption at decentral scale. As a technology of engagement, energy storage thus allows householders to interact with and shape the energy system in new ways. Most of the storage modes allow prosumers with battery systems to generate not only use value (by self-consumption of stored energy), but also exchange value (by sharing and trading energy and providing grid services) [29]. Energy storage thereby leads to more options for prosumers about what they want to do with their self-generated energy and with whom.

When storage affords energy practices in which self-produced energy gets exchange value, an important question is how prosumers will relate to this. Two diverging storylines now get connected to this exchange value: the first presents self-produced energy as a potential source of revenue for householders (energy as commodity), and the second emphasizes the sharing of surplus energy with other households (energy as (common pool) resource). Future social scientific research could follow up on these storylines and analyze the "moral economies" -or in other words moral and ethical questions about the production, distribution and exchange of energy- that emerge around this newly unlocked exchange value.

In examining the ways in which the new energy practices are organized in storage modes, our framework challenges the notion of active and aware citizens owning and operating their own household or community batteries. On the one hand, energy storage enables householders to become more autonomous from conventional suppliers and to enter new exchange relationships with other householders and the energy system. On the other hand, they face new dependencies on intermediaries and opaque information technologies. As long as householders believe that aggregators and algorithms act in their interest, they may not consider this a problem. Our analysis showed, however, similar to Parra et al. [12]), that the real-world applications of energy storage are still very much provider-driven. For existing community groups, it is difficult to initiate storage projects because in most countries legal limitations and complexities block communities from supplying their own energy to its members, or to organize the distribution of energy. In this context of provider-driven storage products and services, the question for householders is if they trust it is their aspirations and interests that are taken into account.

It is with regard to this potential for alternative forms of organizing energy production and consumption that we can identify policy implications. To foster storage modes that take into account a wider range of (future) interests and aspirations of householders and communities, and enable diverse forms of energy citizenship, governments could develop policies to actively support experimentation with social organization. An example of this is the Dutch 'Experimentenregeling' which provides energy cooperatives regulatory lenience to experiment with generating, supplying and distributing energy in their own local network. At the same time, studies have shown that such community-based models face difficulties due to financial, legal, social and technical restrictions and complexities surrounding energy storage and engaging with governance circles [2,17]. Beyond regulatory leniency, two other requirements for enabling experimentation include elimination of some of the financial risks and uncertainties in order to embolden communities as initiators of pilot projects, and secondly,

professional facilitation of householders and communities to enable them to articulate their interests and ambitions vis-à-vis intermediaries. The emergence of prosumer platforms too offers opportunities for co-creation by citizens. Prosumer platforms could be developed or adapted together with local or virtual energy communities to ensure that energy exchange takes place based on valuations of energy and distribution of benefits and costs that the community favors. Opening up spaces for communities to initiate and develop energy storage projects may prevent that some emerging modes become marginalized too soon, and prevent lock-in situations in which existing power relations between providers and householders are reproduced. Recognizing that energy storage (as technology of engagement) offers prosumers enticing—and sometimes conflicting—perspectives on greater energy autonomy and self-sufficiency as well as on greater systems integration, it is important to provide space for experimentation with (mixed modes of) energy storage that both empower householders and communities in the pursuit of their own sustainability aspirations and serve the needs of emerging renewable energy-based energy systems. Providers and policy makers need to recognize that the 'storage revolution' should not just be seen in technical or economic terms, but also as an experiment with multiple new ways of relating to energy and new forms of social organization of energy production and consumption.

**Author Contributions:** Conceptualization, S.K., R.S. and N.V.; methodology, S.K., R.S. and N.V.; validation, S.K., R.S. and N.V.; investigation, S.K., R.S. and N.V.; data curation, S.K., R.S. and N.V.; writing—original draft preparation, S.K., R.S., and N.V.; writing—review and editing, S.K. and R.S.; visualization, R.S.; project administration, S.K.

**Funding:** This research was funded by the Netherlands Organisation for Scientific Research NWO, grant number 408-013-3.

**Acknowledgments:** We are grateful to the DEMAND Center at Lancaster University for co-funding and hosting all three of us to do empirical work in the United Kingdom between April and June 2016. Thanks also to Walter Fraanje for assisting with the empirical work in Germany, and to Marten Boekelo for conducting the interviews in the City-zen project.

**Conflicts of Interest:** The authors declare no conflict of interest. The funders had no role in the design of the study; in the collection, analyses, or interpretation of data; in the writing of the manuscript, or in the decision to publish the results.

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
