# Peer review of "Technologies of Engagement: How Battery Storage Technologies Shape Householder Participation in Energy Transitions"

_energies, doi:10.3390/en12224384_

Round 1

Reviewer 1 Report

It is a good qualitative paper on the study of energy storage as a ‘technology of engagement’.

It is well presented. Just a comment for a minor revision:

A brief discussion about the benefits and challenges of the use of batteries should be added in relation also to other storage technologies. About this, please see:

-  Castellani, B., Morini, E., Nastasi, B., Nicolini, A., Rossi, F. Small-scale compressed air energy storage application for renewable energy integration in a listed building (2018) Energies, 11 (7), art. no. 1921, .

Author Response

We thank reviewer 1 for their time to review our article. The reviewer suggests adding ‘a brief discussion about the benefits and challenges of the use of batteries should be added in relation also to other storage technologies’. We looked up the suggested article, but consider such a discussion as falling outside the scope of our article. The reason for this is that our article is not about the performance of different storage technologies, but about how consumers can interact with home batteries (electricity storage), and through that play a role in the energy system. Hence, the focus is on social aspects, not on technological performance. In our introduction, we make this specific focus explicit with the following sentence (line 65): ‘In this article our aim is to explore potential ways in which home batteries can enable householders to become engaged in the transition towards low-carbon energy systems.

Reviewer 2 Report

This article addresses a topical issue and analyses different modes of energy storage. It identifies five different modes in which technologies and social aspects are combined in different ways. This approach is to be appreciated because in principle it allows a more differentiated understanding of societal consequences.

Although the individual modes are very well worked out (chapter 3), they play only a minor role in the discussion of the results (chapter 4). And it remains unclear why this is the case. Either the different modes differ too little in terms of their effects or the authors do not want to go into these differences in more detail. However, the paper should make it clear why the five modes are no longer explicitly referred to in the discussion section.

The article should also go into more detail on the key concepts on which the analysis is based. What do the concepts of "participation in energy transitions" or "material participation" actually mean in this context and how can these concepts be operationalized and empirically investigated? This remains somewhat vague in the current text. Thus it remains unclear whether and to what extent the different modes of battery use actually lead to a change in social practices and how such changes relate to the question of participation in the transformation of the energy system. Do new everyday practices automatically mean a stronger commitment to transition processes or are households essentially concerned with the (assumed but not realised) economic optimisation of their own electricity production?

Another point relates to the role of suppliers in this emerging market. Although the results indicate that the influence (participation) of households on the energy transition remains very limited, the authors tend to adopt statements from the supplier side (interviews) rather uncritically. Current studies from Germany tend to indicate that the burdens here are very unevenly distributed. Private households largely take the risks of such an acquisition (costs, lifetime, operational failure, uncertain future development of prices, etc.) while suppliers profit from clever framed business cases. Interviews with critics and/or households who deliberately decided not to purchase batteries would have shed more light on this perspective. This is clearly missing in this paper.

The suggestion that more space is needed for experiments is justified and important. All the more striking is the paper's shortcoming that no interviews were conducted with collective storage models operated by communities. However, the reference to more space for experimental settings should be supplemented by concrete requirements, as far as these can be derived from the available results.

Despite all the critical comments above, this is already an interesting, well-written paper. I am looking forward to a revised version.

Author Response

We would like to thank reviewer 2 for the thorough reading of our paper and the helpful suggestions. We will explain point by point how we dealt with the comments

-The reviewer asks us to make clear why the five modes are no longer explicitly referred to in the discussion. In relation to this questions and the comments of reviewer 3 on restructuring the different sections, we decided to move Table 1 to the discussion, as well as the section ‘comparing the five storage modes’ that first belonged to the empirical descriptions of the modes. We feel these sections present a discussion of the different modes. In addition, a sentence is added explaining why the remainder of the discussion focuses on two overarching effects.

-The reviewer also asks for more details on the key concepts. We added an explanation of how we understand the two concepts of ‘participation in energy transitions’ and ‘material participation’ in the section ‘storage modes and new energy practices’

-We agree with the comment that new everyday practices do not automatically mean a stronger commitment to transition processes. The aim of our article, however, is not to systematically assess whether this is the case. Rather, taken the exploratory character of our research, our aim is to conceptualise and identify potential ways in which the use of batteries can foster engagement. To make this clear to the readers, we explained this in the final sentence of the methods section.

-The reviewer writes that current studies indicate that business models for battery storage in households are uneven in their distribution of costs and benefits. We have not been able to find recent studies which about these concerns and would greatly appreciate it if the reviewer could recommend us literature on this subject.

-We agree that interviews with collective storage models operated by communities would have been useful and interesting. However, such models do not yet exist in the Netherlands. The community storage systems that do exist are owned and/or operated by energy system actors.

-Beyond regulatory leniency, two further requirements for experimental space have been added to the Conclusion section: reducing financial uncertainties and professional facilitation of householders and communities.

Reviewer 3 Report

Thank you for the opportunity to review this article. Idea of the article is interesting, but before publication some essential editing is needed. Main notes are below:
• Please clearly mention the main findings of the study and the research method used in the abstract.
• I would recommend you, that the part of the paper “Renewable energy technologies as technologies of engagement” be part 2, not the part of Introduction. The literature review in the part of the paper “Renewable energy technologies as technologies of engagement” should be extended.
• The part of the paper “Storage modes and new energy practices for householders” should be part 3 or merged whit part 2. In this part explanations of Storage modes from the results part should be added. The part of the Results should be dedicated to results of the research, not for theoretical information.
• The results of the interview should be systematized. I suggest you try to show the results in tables.
• Unclear sentences: “Literature in science and technology studies (STS) views technologies not just as material objects, but argues that the social and the technical are co-dependent and co-evolving” (line 78-79); “Renewable energy technologies too have been approached as configurations of the technical and the social” (line 80)
• Line 81 reference is missing.
• Figures and Tables should be adjusted based on requirements of the journal.
• Figure 1 is unclear. Does it really show research approach?

Author Response

We would like to thank the reviewer for the helpful suggestions on the structure of our paper. Below, we explain point by point how we dealt with the comments.

-we rewrote the abstract to make clearer what our qualitative results are.

-We created a new part 2 that now covers the theoretical framework of our paper. The reviewer also asks for an extension of the literature review. Yet, the reviewer does not provide reasons for why the current literature review could be considered insufficient, or examples of what he or she considers relevant debates and literature that we have overlooked. Therefore, we decided to extend the literature review by better explaining our use of the key concepts of ‘material participation’ and ‘participation in energy transitions’ as reviewer 3 suggested.  

-We added the section “Storage modes and new energy practices for householders” to part 2, because we agree with the reviewer that this is part of the theoretical framework. The reviewer also suggests to add parts from the results to this section, namely the explanation of the different storage modes. However, we consider the explanation of the modes to be part of the results, because the use of analytical framework (figure 1) of part 2 enabled us to identify these five modes from the empirical material. To make this clearer, we also moved Table 1 to the discussion, to show that the table is the result of our analysis, rather than it preceding the analysis.

-The reviewer asks for a systematization of interview results, but does not provide suggestions on how these results should be systematized, or reasons for why our current systematization would be lacking. Therefore, we would like to explain to approach we took: Our research is an exploratory qualitative research of which the aim is to understand how householders can engage with energy storage and through that participate in the energy transition. Our approach to systematizing the results has been to derive from all the interviews a number of different ways in which householders can engage with storage. The systematization of the interview results thus consists of grouping the findings into the five different modes.

-We added some lines to explain sentences the reviewer found unclear, added the reference for line 81

-We checked the author guidelines again and adapted table 1. For the figures we could not find any additional requirements. We also added a more detailed description of Figure 1, namely that it is our ‘analytical framework for identifying and analysing storage modes’.

Round 2

Reviewer 2 Report

Below you will find some studies from Germany that critically examine PV storage systems in the private sector.

Graulich, K., Bauknecht, D., Heinemann, C., Hilbert, I., Vogel, M., Seifried, D. und Albert-Seifried, S., 2018, Einsatz und Wirtschaftlichkeit von Photovoltaik-Batteriespeichern in Kombination mit Stromsparen, Ergebnisse aus dem BMBF-geförderten Verbundprojekt BuergEn ‘Perspektiven der Bürgerbeteiligung an der Energiewende unter Berücksichtigung von Verteilungsfragen’; Endbericht Teilprojekt 1, Modul 4.1, im Auftrag von: BMF: Öko-Institut e.V.

Figgener, J., Haberschusz, D., Kairies, K.-P., Wessels, O., Tepe, B. und Sauer, D. U., 2018, Wissenschaftliches Mess- und Evaluierungsprogramm Solarstromspeicher 2.0, Jahresbericht 2018, Aachen: Institut für Stromrichtertechnik und Elektrische Antriebe RWTH Aachen. http://www.speichermonitoring.de/fileadmin/user_upload/Speichermonitoring_Jahresbericht_2018_ISEA _RWTH_Aachen.pdf.

Moshövel, J., Magnor, D., Sauer, D. U., Gährs, S., Bost, M., Hirschl, B., Cramer, M., Özalay, B., Matrose, C., Müller, C. und Schnettler, A., 2015, Analyse des wirtschaftlichen, technischen und ökologischen Nutzens von PV-Speichern, Gemeinsamer Ergebnisbericht für das Projekt PV-Nutzen, im Auftrag von: BWE.

Schill, W.-P., Zerrahn, A., Kunz, F. und Kemfert, C., 2017, Dezentrale Eigenstromversorgung mit Solarenergie und Batteriespeichern: Systemorientierung erforderlich, DIW Wochenbericht (12).

Author Response

We would like to thank the reviewer for his/her time to read the revised version of our manuscript. We would also like to thank the reviewer for the literature suggestions. Since all suggestions are in German and we only posses basic German language skills, we faced difficulties reading the literature.  However, we have added at the end of section 5.3 of a reference to a 2018 UK study on (alternative) business models of domestic storage which addresses concerns about fairness. We also added a critical note to the discussion on new collectives. We hope this is appreciated.

Reviewer 3 Report

Authors have improved the paper according to my observations.

Author Response

We would like to thank the reviewer for his/her time to read the revised version and are happy to hear that we managed to improve the paper.